# Cerebrospinal Fluid Metabolome in Parkinson’s Disease and Multiple System Atrophy

**DOI:** 10.3390/ijms23031879

**Published:** 2022-02-07

**Authors:** Do Hyeon Kwon, Ji Su Hwang, Seok Gi Kim, Yong Eun Jang, Tae Hwan Shin, Gwang Lee

**Affiliations:** 1Department of Molecular Science and Technology, Ajou University, Suwon 16499, Korea; dohyeon248@ajou.ac.kr (D.H.K.); js3004@ajou.ac.kr (J.S.H.); rlatjrrl9977@ajou.ac.kr (S.G.K.); jye120@ajou.ac.kr (Y.E.J.); 2Department of Physiology, Ajou University School of Medicine, Suwon 16499, Korea

**Keywords:** cerebrospinal fluid, integrated omics, machine learning, multiple system atrophy, metabolomics, Parkinson’s disease

## Abstract

Parkinson’s disease (PD) and multiple system atrophy (MSA) belong to the neurodegenerative group of synucleinopathies; differential diagnosis between PD and MSA is difficult, especially at early stages, owing to their clinical and biological similarities. Thus, there is a pressing need to identify metabolic biomarkers for these diseases. The metabolic profile of the cerebrospinal fluid (CSF) is reported to be altered in PD and MSA; however, the altered metabolites remain unclear. We created a single network with altered metabolites in PD and MSA based on the literature and assessed biological functions, including metabolic disorders of the nervous system, inflammation, concentration of ATP, and neurological disorder, through bioinformatics methods. Our in-silico prediction-based metabolic networks are consistent with Parkinsonism events. Although metabolomics approaches provide a more quantitative understanding of biochemical events underlying the symptoms of PD and MSA, limitations persist in covering molecules related to neurodegenerative disease pathways. Thus, omics data, such as proteomics and microRNA, help understand the altered metabolomes mechanism. In particular, integrated omics and machine learning approaches will be helpful to elucidate the pathological mechanisms of PD and MSA. This review discusses the altered metabolites between PD and MSA in the CSF and omics approaches to discover diagnostic biomarkers.

## 1. Introduction

Parkinson’s disease (PD) and multiple system atrophy (MSA) are synucleinopathies, which develop by the transcellular propagation of α-synuclein in neurons and glial cells, causing inclusion bodies such as Lewy bodies (leading to Lewy body disease such as PD) and Papp–Lantos bodies (leading to Papp–Lantos body diseases such as MSA) [1,2]. Owing to the many clinical similarities between the diseases, differential diagnosis of PD from MSA is difficult. Because disease progression and response to classic dopaminergic (DAergic) treatment is different in PD and MSA, accurate diagnosis is important at the early stage of disease for suitable treatment.

Metabolites profiling of the neurodegenerative brain may facilitate biomarker discovery for diagnosis because metabolite levels are altered in pathological conditions and act as biochemical and physiological indicators [3]. Thus, metabolomics approaches have led to the discovery of new metabolic biomarkers for the diagnosis of complicated symptoms [4]. In particular, secondary metabolites are used as biomarkers for the early diagnosis of diseases such as type 2 diabetes [5], COVID-19 [6], and neurodegenerative diseases [7,8,9]. Cerebrospinal fluid (CSF) is the body fluid that surrounds the brain; it is, therefore, the most promising body fluid for identifying biomarkers for neurodegenerative diseases. In addition, metabolites in CSF reflect the pathophysiological conditions in PD and MSA well and provide biomarkers; however, the literature is incomplete and controversial [10]. Classically, metabolomics determines the presence or absence of metabolites and their concentration. To analyze precise biological matrices, concentration ratios and profiling are applied for the extraction of precise phenotypes based on the presence or absence of metabolites and their concentration, which is called metabotypes [11]. Therefore, the CSF metabolome can help distinguish PD and MSA [8]. Thus, metabolomics is a powerful approach to understand the phenotypes of PD and MSA to identify clinically relevant biomarkers. Several proteins have been reported to be more abundant and uniquely identified in CSF than in serum [12]. Thus, CSF has been the target biological fluid as a biomarker for metabolites, microRNAs, and proteins in PD and MSA [8,13,14,15,16].

Although the metabolomics approach is useful, the integrated omics approach provides precise and sensitive information by integrating delicate biological information. For example, metabolomics alone failed to detect delicate nanotoxicity using traditional assay methods [17], while assessment using integrated omics with metabolomics and transcriptomics has been shown to provide a more sensitive and detailed toxicological evaluation of cellular responses to identify novel nanotoxicological biomarkers [18]. Moreover, several studies have suggested the analysis of the proteome [16,19,20,21] and microRNA [13,22] in the CSF of patients with PD and/or MSA. Therefore, integrated omics with metabolomics and proteomics and/or microRNA analysis may contribute to more precise evaluations based on symptoms between PD and MSA in the future. In this review, we discuss the following: (i) characteristic differences between PD and MSA, (ii) metabolomics approaches for PD and MSA, and (iii) the use of proteomics and microRNA data for discovering specific biomarkers for PD and MSA.

## 2. Characteristic Differences between PD and MSA

PD is the second most common neurodegenerative disorder and is characterized by progressive loss of DAergic neurons in the substantia nigra pars compacta [23,24]. Epidemiologic studies have shown the prevalence of PD to be 1–2 per 1000 people in the world and 1% of the population over 65 years [25]. The motor symptoms of PD include tremor, rigidity, and bradykinesia, and non-motor symptoms include fatigue, cognitive impairment, low blood pressure, sleep cycle dysregulation, autonomic dysregulation, and dementia [26,27].

Parkinsonism is defined as a clinical syndrome that causes movement problems in patients with PD. Even though there are various categories of parkinsonism, the most common atypical form of parkinsonism, MSA (also referred to as Shy-Drager syndrome), is a progressive sporadic adult-onset neurodegenerative disorder caused by neuronal cell loss and gliosis in specific areas of the brain, such as the inferior olivary nuclei, spinal cord, pons, ganglia, and cerebellum, causing rigidity, postural instability, and bradykinesia, which are similar to the motor symptoms of PD [28]. The prevalence of MSA is estimated to be 3–5 per 100,000 people, increasing to about 8 per 100,000 people older than 40 years [29]. Its life expectancy is 6-10 years, as it progresses relentlessly after diagnosis [30]. Although it is well known that PD and MSA have different underlying mechanisms, it is difficult to distinguish between the two based on clinical signs and symptoms due to overlapping symptoms [31,32].

Prion-like propagation of α-synuclein is pathophysiologically important indicator in PD and MSA [33], and the misfolded α-synuclein, propagation seed, can spread along neuronal pathways [33]. Especially, α-synuclein pathological template seeding can start in the peripheral nervous system and retrogradely propagate to the brain [34,35]. The existence of diverse so-called α-synuclein “strains” is most likely responsible for the clinical heterogeneity among PD and related synucleinopathies [36]. Indeed, it has been shown that it is possible to detect and discriminate between samples of CSF from patients with PD and samples of CSF from patients with MSA, with a very high sensitivity [37,38].

To understand and distinguish the two diseases, clinical studies as well as in vitro and in vivo models have been developed. PD models have been generated by targeting DAergic neurons. Methyl-4-phenyl-1,2,3,6-tetrahydropyridine (MPTP) is a neurotoxin that induces parkinsonism because it selectively kills DAergic neurons [39]. MPTP passes through the BBB and is metabolized to MPDP^+^ by monoamine oxidase B in astrocytes and further oxidized to form MPP^+^. MPP^+^ is taken up through the dopamine transporter, damages mitochondrial complex I, induces oxidative stress and kills DAergic neurons [40]. 6-hydrooxidopamine (6-OHDA) is a catecholaminergic neurotoxin used in PD research. 6-OHDA forms free radicals, inhibits the activities of mitochondrial I and IV complexes, induces death of DAergic neurons, and causes behavioral disorders in Parkinson’s disease [41]. MG-132 is a proteasome inhibitor that inhibits the activity of the ubiquitin proteasome system and causes the accumulation of ubiquitinated proteins in animal models. Inhibition of the proteasome function activates caspase-3 and induces DAergic neuronal death [42]. Although mature oligodendrocytes do not express α-synuclein, glial cytoplasmic inclusions (GCIs) composed of α-synuclein are found in MSA, indicating that α-synuclein is involved in MSA pathology, and to mimic MSA in several studies, transgenic mice have been established to overexpress α-synuclein in oligodendrocytes [28]. 2′,3′-cyclic nucleotide 3-phosphodiesterase (*CNP*)-hαSyn transgenic mice overexpresses α-synuclein through the murine *CNP* promoter of M2 mice and are used as MSA mouse model [43]. Accumulated α-synuclein is accompanied by significant demyelination, motor deficits, and dystrophic neurites in spinal cord motor neurons and pyramidal tracts in a transgenic mouse model. The proteolipid protein (*PLP*)-hαSyn transgenic mice model was created to overexpress α-synuclein in oligodendrocytes using the *PLP* promoter [44]. In this model, α-synuclein is hyperphosphorylated at the ser129 site, contributing to aggregation and formation of GCI-like inclusions. The myelin basic protein (*MBP*)-hαSyn transgenic mouse model was established to overexpress α-synuclein using the *MBP* promoter [45]. In the *MBP* model, abnormal aggregation of α-synuclein occurred in the cerebellum, neocortex, brainstem, and ganglia. Aggregation of α-synuclein induces myelin loss and neurodegeneration. MSA models have also been generated using chemical drugs *in vivo*. Injection of MPP^+^, a mitochondrial complex I inhibitor, and 3-nitropropionic acid (3-NP), a succinate dehydrogenase inhibitor, can induce MSA and neuronal degeneration by inducing mitochondrial dysfunction [46,47]. MSA model mice are also generated by the induction of striatonigral degeneration by administering 6-OHDA into the medial forebrain bundle and quinolinic acid [48].

## 3. Metabolomics Approaches for PD and MSA in CSF

Metabolomics approaches offer a more comprehensive understanding of biochemical events in the development of PD and MSA than conventional methods. In particular, quantitative metabolomics reflects metabolic phenotypes (metabotypes) and unveils indicators of pathological conditions [49,50]. Thus, metabolomics approaches help identify new biomarkers for the diagnosis of PD and MSA. In this section, we discuss the analysis of metabolites, the CSF metabolome differences between PD and MSA, and the application of the metabotypes.

### 3.1. Analysis of Metabolites

Metabolites are intermediates or end products of metabolism and are normally used for low molecular weight metabolites (less than 1000 Da) and catalogued approximately 2500 metabolites in humans [51]. Metabolomics is the comprehensive study of metabolic changes and the large-scale study of metabolites in human cells, tissues, and organs, including fluids [52]. Compared to other omics technologies, metabolomics directly reflects biological processes, such as the regulation of enzyme activity, cellular signaling, energy metabolism and conversion, and interactions with other organisms [53,54,55]. Primary metabolites such as amino acids, fatty acids, organic acids, carbohydrates, and vitamins are essential for growth, development, and reproduction, and are required for maintaining the physiological functions of the human body. Secondary metabolites such as polyamines, catecholamines, hormones, antibiotics, and steroids are derivatives of primary metabolites and are formed during the stationary phase of growth. Metabolites are involved in various biological functions and play important roles in neurodegenerative diseases such as Alzheimer’s, Parkinson’s, Huntington’s disease, and multiple sclerosis [56].

The two most common analytical instruments used in metabolic profiling are nuclear magnetic resonance (NMR) and mass spectrometry (MS). First, although NMR possesses a relatively low sensitivity (>1 nmol) and resolution, and low detectable metabolites compared to MS, NMR possesses relatively high reproducibility, minimal sample preparation requirement, and low cost per sample [57,58]. To overcome the low sensitivity of NMR spectroscopy, cryogenic NMR spectroscopy has been developed for metabonomic studies [59]. Compared to NMR, MS possesses high sensitivity and accuracy for the detection of molecules and is commonly used in combination with liquid chromatography (LC), gas chromatography (GS), and capillary electrophoresis (CE) for separation [60].

### 3.2. Metabolomic Differences in the CSF in PD and MSA

As metabolites are important biochemical and physiological indicators of different pathophysiological conditions in the brain, altered metabolomes are good candidate biomarkers for PD and MSA. For example, the level of the long-chain omega-3 fatty acid, eicosapentaenoic acid (EPA), which is associated with key anti-inflammatory function, was increased in the CSF of patients with PD and MSA compared to those of the controls [61]. The level of a four-carbon linear chain diamine, putrescine (1,4-diaminobutane), was increased in the CSF of patients with PD compared to those of control and MSA patients [8]. Recently, it was reported that the levels of 3-methoxy-4-hydroxyphenylglycol (MHPG) and norepinephrine (NE) were decreased in the CSF of patients with PD and MSA compared to those of the controls [10]. However, this report is different from another report that showed low CSF NE in the PD and MSA group and reduced MHPG in only the MSA group compared to the control group [7].

Compared to MSA, over two times of PD metabolites have been reported to be related to CSF. Among them, similarly categorized metabolites are summarized in the CSF of PD and MSA compared with the control (Table 1). The criteria of upregulation and downregulation is statistically altered, and controversial data are excluded. Besides these, many altered metabolites in the CSF of patients with PD compared with controls have been reported. The levels of trans-4-hydroxyproline [62], α-N-phenylacetyl-L-glutamine [63], (+)-gamma-hydroxy-L-homoarginine [63], p-cresol sulfate [64], decanoic acid [64], 10-hydroxydecanoic acid [64], dihomo-γ-linolenic acid [64], and diacylglycerol [65] have been shown to be increased, while those of betaine [63], dimethylglycine [63], α-aminobutyric acid [66], ornithine [66], lysine [66], and histidine [66] have been shown to be decreased.

There are limitations to the interpretation of altered metabolites and neurodegenerative events. We used a web-based bioinformatics software, Ingenuity Pathway Analysis (IPA, http://www.ingenuity.com, accessed on 16 January 2022) to analyze the biological phenomena occurring in PD and MSA using the CSF metabolome [67]. To visualize the network of metabolites, integrated networks were constructed based on Table 1, as shown in Figure 1 (PD vs. control) and Figure 2 (MSA vs. control). These integrated metabolic networks revealed that altered metabolomes were related to four biological functions, namely, metabolic disorders of the nervous system, inflammation, concentration of ATP, and neurological disorders in PD (Figure 1A). In-silico prediction based on metabolic networks is consistent with parkinsonism events such as metabolic disorders of the nervous system [68], inflammation [69], the concentration of ATP [70], and neurodegenerative disease [71] in PD patients (Figure 1B). Due to the small size of metabolites, there was no significant difference in the functional analysis of the metabolomic network of CSF between PD and MSA using in-silico prediction (Figure 1B and Figure 2B). In the future, it might be possible to distinguish these if more data on metabolites and other omics data, such as proteomics and microRNA omics, is accumulated.

Polyamines are low-molecular-weight organic molecules containing two or more amino groups [80]. Polycationic polyamines bind to various negatively charged biological molecules and play a role in the regulation of biological processes, including cell growth, survival, and proliferation [81]. In neurodegeneration related to protein misfolding, cytoplasmic inclusions are found in synucleinopathies. The inclusions mainly consist of ubiquitin, α-synuclein, and synphilin-1 in synucleinopathies [82,83,84]; the isoelectric points (pI) of ubiquitin, α-synuclein, and synphilin-1 are 6.79, 4.76, and 5.96, respectively [85]. Positively charged polyamines have been shown to interact with negatively charged proteins that mainly occur in the inclusion bodies as the pI values of these proteins are lower than the cytoplasmic pH (ca 7.0–7.4) [86]. In particular, α-synuclein aggregation has been reported to be boosted by the presence of cationic molecules [87,88]; another study suggested tri- and tetra-cationic polyamines, such as spermine and spermidine, incorporated more frequently to bind α-synuclein proteins compared to bi-cationic putrescine [89]. Moreover, cellular putrescine of SH-SY5Y cells (human neuroblastoma cell line) was higher than that of spermine and spermidine in the in vitro inclusion formation conditions based on polyamine profiling with GC-MS and transcriptomic analysis with polyamine-related enzymes [85]. In particular, this report suggests that the accumulation of putrescine occurs because the rate of conversion of putrescine to spermidine by spermidine synthase 1 (SRM1) is slower than the rate of spermidine reconversion to putrescine by spermidine/spermine acetyltransferase 1 (SAT1) and polyamine oxidase (PAOX). Interestingly, putrescine concentration was found to be significantly higher in the CSF of the PD group than in the control and MSA groups [8] and was suggested to be responsible for restricted neuronal degeneration in the substantia nigra in PD, as MSA involves much more progressive and widespread neurodegeneration in the brain. To date, there are no acceptable biomarkers for the differential diagnosis between PD and MSA. Therefore, large-scale metabolomics studies investigating the changes in the CSF of PD and MSA may be useful for identifying specific biomarkers for the accurate diagnosis of PD and MSA. In particular, for definitive diagnoses of PD and MSA, more sophisticated metabolic studies in the brain, leading to the development of accurate and rapid analytical diagnostic biomarkers are needed.

### 3.3. Application of Metabotypes

A major challenge for 21st-century medicine has been defining the relationships between genetic variation and environmental triggers of diseases. Systems biology, supported by genomics, proteomics, bioinformatics, and metabolomics, has emerged as a potential tool for understanding complex biological relationships [90]. Even though metabolome data have comparatively low coverage compared to other omics data (genome, transcriptome, proteome), it is quantitative data that is a good reflection of the phenotype, owing to the sensitivity of internal and external signals in physiological homeostasis. Traditionally, metabolomics studies have focused on the presence or absence of metabolites and their concentrations. More precise information for the analysis of physiological status can be extracted from concentration ratios and profiling of sensitive metabolites. In the literature, metabolic phenotypes (metabotypes or chemotypes) are often defined based on four metabolic variability criteria, namely, the presence or absence of metabolites, their concentration levels, ratios, and profiles of metabolites in biological matrices [11,91]. Paik et al. reported that, in the CSF of PD group, as compared with the CSF of MSA group, the ratio of putrescine spermidine-1 was significantly increased and profiling pattern was readily distinguishable based on the presence or absence of metabolites and their concentration [8]. For metabotyping, metabolic profiling should be performed for sensitive metabolites. Thereby, metabotyping by analyzing metabolic profiles can provide a chance to identify new biomarkers for disease diagnosis [92].

## 4. Proteomics and MicroRNA Analysis in PD and MSA

Although metabolomics approaches provide a more comprehensive understanding of biochemical events involved in the symptoms of PD and MSA, there are limitations to cover molecules related to neurodegenerative pathways. Thus, other omics approaches help identify new biomarkers for the diagnosis of PD and MSA. In this section, we discuss proteomics, microRNAs, and integrated omics analysis to understand the biochemical mechanisms of PD and MSA.

### 4.1. Proteomics

Proteomics has been applied for the investigation of neurodegenerative diseases [93] because it permits understanding molecular processes, compositions, sizes, and charges of related proteins [55]. There have been analytical limitations with the use of CSF samples of patients with neurodegenerative diseases using classical analytical techniques, such as Western blotting and two-dimensional (2D) gel electrophoresis based on pI. Changes in the protein levels have been analyzed in the CSF of PD and MSA patients against healthy individuals. In the proteome profile of PD, levels of 30 proteins were increased and levels of 40 proteins were decreased in a total of 110 proteins [16,19,20,21,73,94,95]. In addition, the levels of 28 proteins were increased and levels of 20 proteins were decreased in a total of 73 proteins detected in MSA [16,20,73,95]. Of the differentially expressed proteins, the expression of seven proteins was increased and expression of nine proteins was decreased in PD and MSA compared to healthy controls [16,20,73,95]. Of the proteins increased in both PD and MSA, the increment of insulin-like growth factor 2 (IGF2), glial fibrillary acidic protein (GFAP), eotaxin, and interleukin 10 (IL10) was more pronounced in MSA and that of seizure 6-like protein 2 (SEZ6L2), 45 kDa calcium-binding protein (SDF4), and granulocyte colony-stimulating factor (G-CSF) was more pronounced in PD. Of the proteins decreased in both PD and MSA, the decrement of thy-1 membrane glycoprotein (THY-1), protocadherin gamma-C5 (PCDHGC5), Parkinson disease protein 7 (PARK7/DJ-1), α-synuclein, epidermal growth factor (EGF), interleukin 9 (IL9) was more pronounced in MSA. Moreover, protein O-linked mannose beta-1,2-N-acetylglucosaminyltransferase 1 (POMGNT1), extracellular matrix protein 1 (ECM1), cadherin-2, secretogranin-2, SLIT and NTRK-like protein 1 (SLITRK1), contactin-1, and interleukin 7 (IL7) were upregulated in PD but downregulated in MSA compared to healthy controls. Neuroblastoma suppressor of tumorigenicity 1 (NBL1), neurofascin, cluster of differentiation (CD44), carboxypeptidase E, ceroid-lipofuscinosis neuronal protein 5 (CLN5), N-acetylmuramoyl-L-alanine amidase (PGLYRP2), myelin basic protein (MBP), tau, and interferon α-2 (IFNA2) have been found to be upregulated in MSA but downregulated in PD [16,20,73,95]. Therefore, proteomics can be one of the approaches used to identify potential biomarkers for PD or MSA.

### 4.2. MicroRNAs

MicroRNAs are a class of small, non-coding RNAs comprising approximately 20 nucleotides [96]; they control post-transcriptional regulation of mRNA and regulate their translation by binding to the 3′UTR. They were first discovered in Caenorhabditis elegans in 1993 and were named “mediators of temporal pattern formation” [97,98]. MicroRNAs have been studied and used as clinical biomarkers in many diseases, including neurodegenerative diseases [99,100]. MicroRNAs regulate protein levels in biological fluids, including cell-free CSF, and have been identified as potential biomarkers in neurodegenerative diseases. Seventeen microRNAs have been found to be more significantly differentially expressed in the CSF of patients with PD than in healthy controls [22]. The level of miR-205 was upregulated, but the level of miR-24 was downregulated in the CSF of PD patients compared to healthy controls. In addition, four microRNAs (miR-19a, miR-19b, miR-24, and miR-34c) were downregulated in the CSF of patients with MSA compared to healthy controls [13]. These reports suggest that microRNAs are good candidates as potential biomarkers for PD and MSA.

### 4.3. Integrated Omics

The metabolic profile is an endpoint of biological metabolism and closely reflects the corresponding phenotype [54,55]. However, metabolites cannot be amplified; therefore, metabolomics has relatively lower coverage than transcriptomics and genomics [101]. To compensate for the weakness of metabolomics, integration of omics technologies is highly recommended, but the development of this strategy is challenging [102]. Integrated omics studies have identified novel biological changes underlying various conditions in vitro [18,85,103,104,105,106,107]. There have been limitations in the analysis of other omics data, except for metabolites, from CSF samples due to technical issues. However, recently, proteome analysis using LC-MS/MS [108] and antibody array [109] have been developed, and quantitative PCR for microRNAs can determine the levels of microRNAs [13]. Such analytical improvement enables researchers to overcome limitations and analyze clinical symptoms comprehensively.

Machine learning is used for handling large-scale datasets, including integrated omics data [105]. Machine learning is divided into supervised learning, unsupervised learning, and reinforcement learning [55,110]. Supervised learning typically includes support vector machine (SVM), K-nearest neighbor (KNN) [111]. Unsupervised learning involves training with unlabeled data and grouping them into similar groups [112]. In the unsupervised learning clustering algorithm, there are K-means, K-medoids, and hierarchical density-based spatial clustering of applications with noise (HDBSCAN) clustering [55,110]. In addition, principal component analysis (PCA) facilitates data analysis by reducing the dimensions of the data distribution [113]. Large-scale multi-omics datasets can be reduced in dimension by performing PCA and organized through a clustering algorithm for unsupervised learning. The application of machine learning algorithms will help reduce biases in the analysis and trimming of integrated omics data, including metabolomic, proteomic, and microRNA data from the CSF of PD and MSA patients [105].

## 5. Summary

We reviewed the literature on metabolite alteration in CSF from PD and MSA patients and analyzed biological functions such as metabolic disorders of the nervous system, inflammation, concentration of ATP, and neurological disorders through bioinformatics methods in PD and MSA and compared it with healthy controls. Our in-silico prediction-based metabolic networks were consistent with Parkinsonism events. Although biological functions are consistent with Parkinsonism events, there is a pressing need for the integration of other omics data, including proteomics and microRNAs. In the future, other omics data that analyze CSF can be integrated, and integrated omics and machine learning approaches may help elucidate the pathological differences between PD and MSA. Thus, these approaches will contribute to elucidate the pathological mechanism and identifying vital biomarkers for the diagnosis of PD and MSA.

## Figures and Tables

**Figure 1 ijms-23-01879-f001:**
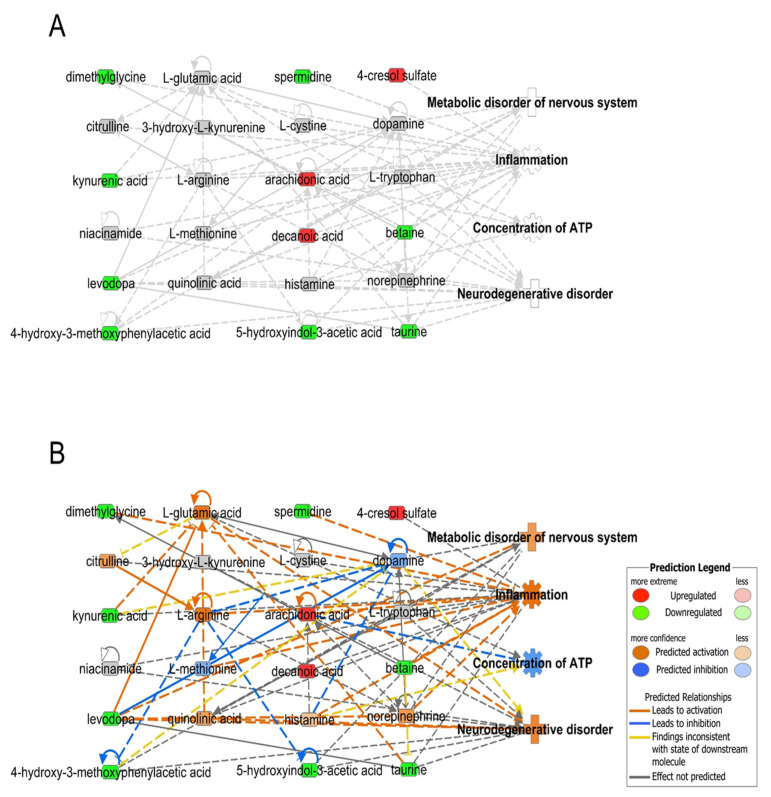
Functional analysis of the metabolomic network of cerebrospinal fluid from Parkinson’s disease patients using the ingenuity pathway analysis (IPA) program. Non-predicted metabolomic network (**A**) and predicted metabolomic network (**B**). Red and green areas indicate upregulated and downregulated metabolites, respectively. Orange and blue areas indicate activation and suppression by IPA prediction, respectively. Prediction legends are provided from IPA (http://www.ingenuity.com, accessed on 16 January 2022).

**Figure 2 ijms-23-01879-f002:**
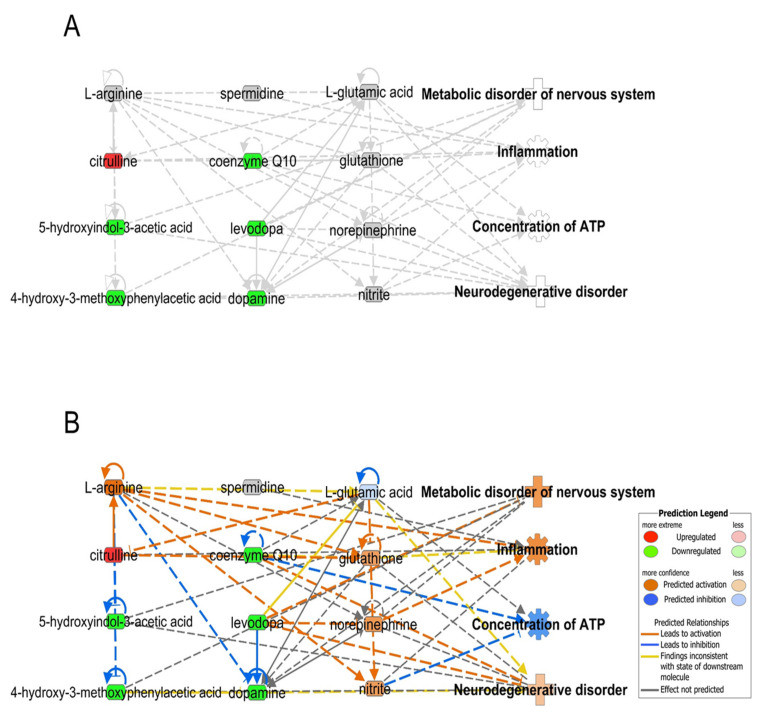
Functional analysis of the metabolomic network of cerebrospinal fluid from multiple system atrophy patients using the ingenuity pathway analysis (IPA) program. Non-predicted metabolomic network (**A**) and predicted metabolomic network (**B**). Red and green areas indicate upregulated and downregulated metabolites, respectively. Orange and blue areas indicate activation and suppression by IPA prediction, respectively. Prediction legends are provided from IPA (http://www.ingenuity.com, accessed on 16 January 2022).

**Table 1 ijms-23-01879-t001:** Altered metabolites in PD and MSA CSF.

PD vs. Control				MSA vs. Control			
**Metabolite**	**Alteration**	Analysis Equipment	Reference	Metabolite	Alteration	Analysis Equipment	Reference
EPA	↑	GC-MS	[61]	EPA	↑	GC-MS	[61]
5-HIAA	↓	Sandwich ELISA	[72]	5-HIAA	↓	Sandwich ELISA	[72]
HVA	↓	HPLC	[73]	HVA	↓	HPLC	[73]
DOPA	↓	LC	[9]	DOPA	↓	LC	[9]
DHPG	↓	LC	[9]	DA	↓	LC	[9]
DOPAC	↓	LC	[9]	DOPAC	↓	LC	[9]
Tyrosine	↑	HPLC	[62]	NE	N.S	LC	[9]
Taurine	↓	HPLC	[74]	DHPG	↓	LC	[9]
KYNA	↓	UPLC	[75]	MHPG	↓	HPLC	[73]
QNA	↑	FT-ICR-MS	[64]	N1-Acetylputrescine	↓	GC-MS	[8]
N1-Acetylcadaverine	↑	GC-MS	[8]	N1-Acetylcadaverine	↓	GC-MS	[8]
Putrescine	↑	GC-MS	[8]	Putrescine	N.S	GC-MS	[8]
Cadaverine	↑	GC-MS	[8]	Cadaverine	↑	GC-MS	[8]
N1-Acetylspermidine	↑	GC-MS	[8]	N1-Acetylspermidine	↓	GC-MS	[8]
N8-Acetylspermidine	↑	GC-MS	[8]	N8-Acetylspermidine	↑	GC-MS	[8]
Spermidine	↓	GC-MS	[8]	Spermidine	N.S	GC-MS	[8]
Arachidonic acid	↑	FT-ICR-MS	[64]	Citrulline	↑	HPLC	[76]
Alanine	↓	Amino acid analyzer	[66]	Arginine	N.S	HPLC	[76]
Valine	↓	Amino acid analyzer	[66]	Glutamate	N.S	HPLC	[76]
Isoleucine	↓	Amino acid analyzer	[66]	Coenzyme Q10	↓	ELISA	[77]
Leucine	↓	Amino acid analyzer	[66]	Glutathione	N.S	HPLC	[78]
Ethanolamine	↓	Amino acid analyzer	[66]	Nitrite	N.S	ELISA	[79]
Nitrate	↓	ELISA	[79]	Nitrate	↓	ELISA	[79]

Abbreviations: EPA, Eicosapentaenoic acid; 5-HIAA, 5-Hydroxyindoleacetic Acid; HVA, homovanillic acid; DOPA, 3,4-dihydroxyphenylalanine; DOPAC, 3,4-dihydroxyphenylacetic acid; DHPG, 3,4-dihydroxyphenylacetic acid; KYNA, kynurenic acid; QNA, quinic acid; DA, dopamine; NE, norepinephrine; MHPG, 3-methoxy-4-hydroxyphenylglycol; N.S, not significant; ↑, up-regulation; ↓, down-regulation.

## Data Availability

The data supporting the findings of this study are available from the corresponding author, upon reasonable request.

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
