# Peer review of "Cerebrospinal Fluid Metabolome in Parkinson’s Disease and Multiple System Atrophy"

_ijms, 2022, doi:10.3390/ijms23031879_

Round 1
Reviewer 1 Report
In the manuscript entitled “Cerebrospinal fluid metabolome in Parkinson's disease and multiple system atrophy” authors have highlighted the distinguishing features of PD and MSA and altered metabolites in the CSF between PD and MSA patients. Furthermore, authors have created a metabolomic network of cerebrospinal fluid from PD and MSA patients using the ingenuity pathway analysis program and analyzed biological functions for instance metabolic disorders of the nervous system, inflammation, concentration of ATP, and neurological disorders. This article is systematically organized with relevant information under appropriate subheadings and emphasized in two figures depicting the integrated metabolic networks of cerebrospinal fluid from PD and MSA separately. The information summarized in this review is going to enrich the existing knowledge to the arena of research on finding of biomarkers for the diagnosis of PD and MSA; hence, this review should be accepted for the publication in the journal of “IJMS”.
Plagiarism percentage - not checked by the reviewer.
Author Response
We appreciate the positive comments of the reviewer.
Reviewer 2 Report
I read with great interest the manuscript by Kwon & colleagues. In
this manuscript the authors review the role of the metabolic profile of
the CSF in PD and MSA pathophysiology. Overall, the manuscript is
well-writen and might be of potential interest for Synucleinopathies
research community.
Nevertheless, I have some concerns that the authors should address in
the revised version of the manuscript:
The authors practically ignored to acknowledge one of the hottest - if
not the hottest - topic in PD and MSA field in recent years, i.e. the
prion-like propagation of misfolded aSyn, that is the pathophysiological
gold-standard of such diseases. Compiling evidence has shown that aSyn
“seeds” can spread along neuronal pathways (PMID: 34361100). Notably,
aSyn pathological template seeding can start in the PNS and retrogradely
propagate to the brain (PMID: 33880502; PMID: 32859276).
The existence of diverse so-called aSyn “strains” is most likely
responsible for the clinical heterogeneity among PD & related
Synucleinopathies (PMID: 26324905). Indeed, it has been
shown that it is possible to detect and discriminate
between samples of CSF from patients diagnosed with PD and samples from
patients with MSA, with a very high sensitivity (PMID: 32025029; PMID: 32341240).
Therefore, the authors should acknowledge all these key studies and
speculate how CSF metabolites might stimulate aSyn seeding and/or
potentially lead to the formation of novel aSyn “strains” in the CSF
that might account for the striking clinical heterogeneity observed in
PD and MSA. '
Round 2
Reviewer 2 Report
The authors have satisfactorily addressed my comments, and I therefore recommend the revised version of the manuscript for publication.